# PROSODYBERT: SELF-SUPERVISED PROSODY REPRESENTATION FOR STYLE-CONTROLLABLE TTS

## ABSTRACT

We propose ProsodyBERT, a self-supervised approach to learning prosody representations from raw audio. Different from most previous work, which uses information bottlenecks to disentangle prosody features from lexical content and speaker information, we perform an offline clustering of speaker-normalized prosody-related features (energy, pitch, etc.) and use the cluster labels as targets for HuBERT-like masked unit prediction. A span boundary loss is also used to capture long-range prosodic information. We demonstrate the effectiveness of ProsodyBERT on a multi-speaker style-controllable text-to-speech (TTS) system, showing that the TTS system trained with ProsodyBERT features generate natural and expressive speech samples, surpassing Fastspeech 2 (which directly models pitch and energy) in subjective human evaluation. In addition, we achieve new state-of-the-art results on the IEMOCAP emotion recognition task by combining our prosody features with HuBERT features, showing that ProsodyBERT is complementary to popular pretrained speech self-supervised models. [1]

## 1 INTRODUCTION

Human speech contains information beyond the associated word sequence. For example, the intonation, stress, rhythm, and tempo of speech carry important cues associated with the speaking style, emotion, and intent. These factors are generally referred to as prosody. Prosodic modeling has been widely investigated in expressive text-to-speech (TTS) synthesis (Valle et al., 2020; Ren et al., 2021; Kenter et al., 2020; Ren et al., 2022) and voice conversion (VC) (Kreuk et al., 2021; Zhou et al., 2022), and has been shown to be important for generating natural and expressive synthesized speech. Prosody is also applied in spoken language understanding tasks by providing information that disambiguates and complements information in the associated word sequence. Examples include parsing (Tran et al., 2018), punctuation prediction (Klejch et al., 2017; Cho et al., 2022), emotion recognition (Rao et al., 2013), and other paralinguistic recognition tasks.

Prosody is traditionally defined in terms of its function in communicating linguistic structure and paralinguistic information and/or in terms of the associated acoustic correlates, which includes fundamental frequency ($F_0$), energy, duration, and other measures associated with vocal effort. In this work, we focus on acoustic correlates, specifically $F_0$ and energy, with duration implicitly encoded via the temporal dynamics. These features have limitations. $F_0$ tracking algorithms are known to be unreliable in some contexts, with pitch halving and doubling errors. Energy is sensitive to recording conditions. $F_0$ and duration depend on speaker and segmental context. Further, $F_0$, energy, and duration are highly inter-dependent, but are often modeled independently, which can lead to unnatural prosody in TTS and limit their usefulness in speech understanding. For these reasons, researchers have been exploring automatic methods for learning alternative representations of prosody.

Automatically learned prosody representations have been proposed for speech synthesis using autoencoders that condition on text and speaker identity, which encourages residual information (assumed to be prosody) to be captured in an information bottleneck (Skerry-Ryan et al., 2018; Wang et al., 2018; Zhang et al., 2019; Qian et al., 2020). These approaches rely on having high-quality speech transcripts, limiting the amount of data that can be used in training and the ability to learn a broadly generalizable representation. Another paradigm of representation learning is self-supervised learning

---

[1]Audio samples are available at: `https://neurtts.github.io/prosodybert_demo/`.

(SSL). SSL models are pretrained on a large amount of unlabeled examples and then finetuned on task-specific data. This paradigm has been particularly successful for natural language processing (Peters et al., 2018; Devlin et al., 2019). Recent speech SSL models like wav2vec 2.0 (Baevski et al., 2020), HuBERT (Hsu et al., 2021), and WavLM (Chen et al., 2022) have been proposed to learn acoustic representations from untranscribed speech. Focusing on the phone level, they achieve good performance on speech recognition and understanding tasks, especially when only a small-amount of task-specific data is available. SSL methods have also been explored for prosody learning in Weston et al. (2021), but the approach requires word time marking, which relies on having human transcripts.

To address the challenge of learning a prosody representation without word transcripts, we propose ProsodyBERT, a self-supervised learning method that disentangles prosody features from speech content and speaker information. Similar to HuBERT (Hsu et al., 2021), we pretrain an SSL model by masked unit prediction. The pseudo labels are given by K-means clustering on speaker-normalized acoustic-prosodic attributes (pitch, energy, and related features), which encourages the model to focus on prosody learning. In addition, inspired by SpanBERT (Joshi et al., 2020), we propose a span boundary loss to encourage the model to better represent long-range prosody information. We also substantially compress the model size and reduce the feature dimensions to make the model easy to use. Similar to prior SSL models, ProsodyBERT is first trained on a large amount of raw speech audio and then adapted to target tasks. Such a design enables ProsodyBERT to learn a rich representation of prosody on massive amounts of untranscribed speech.

Our approach follows that of recent speech synthesis work, which aims to disentangle prosody from lexical content and speaker identity. With this view, acoustic-prosodic features that can be speaker-dependent, such as $F_0$ range, are accounted for in the speaker representation. Disentangling the speech representation into different factors can improve the model's ability to generalize across different conditions and enable zero-shot speaker models for synthesis. While this factoring could be done in different ways (and include other factors), learning a prosody representation that has minimal speaker information is also useful for privacy-sensitive speech processing. Using speaker verification experiments, we show that ProsodyBERT is effective at providing de-identified prosody features.

We demonstrate the effectiveness of pretrained ProsodyBERT features on text-to-speech (TTS) and emotion recognition. During training, we extract ProsodyBERT features from speech and use them as conditional inputs for the TTS decoder. A separate prosody predictor is trained such that it takes text and style as inputs and generates prosody features. During inference, the TTS decoder takes the predicted prosody features as input. Experiments show that the TTS system trained with ProsodyBERT features generates natural and expressive speech, surpassing FastSpeech 2 (Ren et al., 2021) (trained with energy and $F_0$) by a large margin in subjective human evaluation. The expressiveness can be controlled by using speaking-style vectors (learned from multi-style data) in prosody prediction. For the emotion recognition task, we simply concatenate ProsodyBERT features to HuBERT features and use them as the inputs for downstream models. We achieve a new state of the art on IEMOCAP emotion recognition task, showing that ProsodyBERT features are complementary to HuBERT features.

In summary, the key contribution of this work is in the development of a new, low-dimensional representation of prosody that can be learned from untranscribed speech. We demonstrate good speaker de-identification and utility of the new features in both TTS and emotion recognition.

## 2    RELATED WORK

**Modeling Prosody in TTS and VC**    Most prior work on prosody treats prosody feature learning as an auxiliary module for downstream generation tasks. Recent approaches include directly using signal prosody ($F_0$, energy, etc.) (Wan et al., 2019; Valle et al., 2020; Ren et al., 2021; Kenter et al., 2020; Liu et al., 2021; Kharitonov et al., 2022), learning a latent style embedding (Wang et al., 2018; Zhang et al., 2019; Hsu et al., 2019; Sun et al., 2020), learning frame-level or phone-level representation (Du & Yu, 2021; Kreuk et al., 2021), and utilizing reference audios for style (Choi et al., 2020; Yi et al., 2022). Most of these prosody representations rely on task-specific models.

**Learning Prosody Representations**    Prior work on prosody representation learning approaches are mainly based on either information bottleneck or self-supervised learning. Information bottleneck

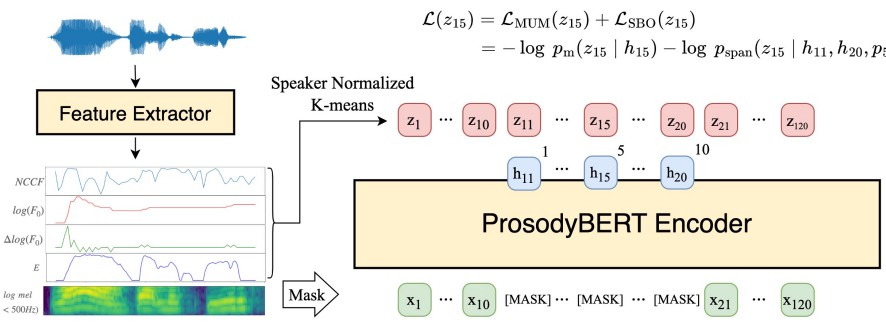

Figure 1: Overview of ProsodyBERT pretraining. Frame-level acoustic-prosodic features (NCCF, $F_0$, energy, deltas) and the log Mel spectrum (bins $< 500Hz$) are first extracted from the audio file. Offline K-means clustering is done on the speaker-normalized acoustic-prosodic features to generate the hidden cluster assignments ($z_{11}, z_{12}, ..., z_{20}$) as the prediction targets. The equation shows the loss terms on the masked frame $x_{15}$. The masked unit modeling (MUM) objective uses the corresponding model outputs $h_{15}$ to predict $z_{15}$. The span boundary objective (SBO) uses the model outputs on the boundary to predict the cluster assignments of every frame in the masked span. Here $z_{15}$ is predicted given $h_{11}, h_{20}$, and the relative position embedding $p_5$.

approaches rely on carefully designed bottlenecks that condition on lexical and speaker information to capture the residual prosody information (Qian et al., 2020; Ren et al., 2022). Other approaches operate more directly on acoustic-prosodic correlates. Weston et al. (2021) learns word-level prosody features from a down-sampled signal by combining a temporal convolutional network and product quantizer with transformer-based contextualization trained using a contrastive task similar to Baevski et al. (2020). However, it relies on word boundaries provided by transcripts to learn word-level prosody representation. Zayats et al. (2019) learn a word-level prosody representation that uses deviations from text-based predictions of $F_0$, energy and duration to compute a prosody representation that is disentangled from lexical cues. Polyak et al. (2021) augments SSL acoustic representation learning with an unsupervised, discrete representation of $F_0$ obtained using VQ-VAE (van den Oord et al., 2017). Our approach differs from all of these methods in that it learns frame-level embeddings from untranscribed speech, leveraging multiple acoustic correlates of prosody.

## 3 PROSODYBERT

An overview of the ProsodyBERT model is given in Figure 1. Our model is based on HuBERT (Hsu et al., 2021) and SpanBERT (Joshi et al., 2020), but is modified to focus on prosody representation learning that disentangles prosody from lexical content and speaker information.

**Notation** Given an audio segment $A$, the feature extractor outputs a sequence of acoustic features $X = (x_1, x_2, \cdots, x_n)$, where each $x_t$ corresponds to a fixed-length frame and $n$ is the number of frames. ProsodyBERT takes $X$ as input and produces a contextualized vector representation for each frame $H = (h_1, h_2, \cdots, h_n)$ that is trained to predict unsupervised prosody units $Z = (z_1, z_2, \cdots, z_n)$. The vectors $h_t$ are the ProsodyBERT outputs.

**Feature Extractor** Given the raw audio segment $A$, the feature extractor first implements loudness normalization on $A$ to 0dB, and then computes frame-based prosodic and spectral features. For prosodic features, we include log fundamental frequency ($\log(F_0)$), Normalized Cross Correlation Function (NCCF), energy, and $\Delta \log(F_0)$. $F_0$ and NCCF are extracted via the Kaldi pitch tracker (Povey et al., 2011); $F_0$ values are interpolated in unvoiced regions. Pitch ($\log(F_0)$) and energy ($E$) are globally normalized on the corpus-level by z-score, and the deltas are computed on the normalized values. No normalization is done on NCCF features. For spectral features, we take the low-frequency bands (first 20 bins, $< 500Hz$) of log Mel spectrum, focusing on the typical range of $F_0$ and filtering out much of the information associated with lexical content. Mean and variance z-score normalization are performed on these low-frequency bins on the utterance level. $\log(F_0)$, $E$ , NCCF, $\Delta \log(F_0)$, and log Mel features are concatenated at each frame as the output $X$.

**Hidden Units for ProsodyBERT**  Inspired by HuBERT (Hsu et al., 2021), we get the frame-level target labels by acoustic unit discovery. Specifically, we train a K-means clustering model on speaker-normalized prosody features ($NCCF, \log(F_0), \Delta \log(F_0), E$) using all the frames in the pretraining corpus. Z-score speaker normalization is done on $E$ and $\log(F_0)$ before clustering to disentangle speaker information. For a speech utterance with acoustic features $X = (x_1, x_2, \cdots, x_n)$, the discovered acoustic units are $Z = (z_1, z_2, \cdots, z_n)$, where $z_t$ is a $K$-class categorical variable.

**Training objectives**  We adopt the same masking mechanism as SpanBERT (Joshi et al., 2020), wav2vec 2.0 (Baevski et al., 2020), and HuBERT (Hsu et al., 2021). Let the input utterance be $X = (x_1, x_2, \cdots, x_n)$. Denote the set of all masked frames as $M \subset X$. Define $\tilde{X}$ as the corrupted input sequence, in which the frames in $M$ are replaced with a special mask embedding. ProsodyBERT has two training objectives. The first is the masked unit modeling (MUM) objective. Let the output hidden states at the $t$-th frame be $h_t$. We compute the masked unit modeling loss of frame $t$ as:

$$\mathcal{L}_{\text{MUM}(z_t)} = -\log\ p_m(z_t \mid h_t). \tag{1}$$

The masked prediction model $p_m$ takes $h_t$ as input and predicts a distribution over target label $z_t$. It is implemented as a single-layer network followed by a softmax.

Previous work mostly relies on alignments to learn the span-level representation of speech (Weston et al., 2021; Hu et al., 2021). In contrast, our self-supervised method only relies on raw audio. Inspired by SpanBERT (Joshi et al., 2020), we add a span boundary loss to encourage ProsodyBERT to learn a long-range prosodic representation. Given a masked span ($x_s, \cdots, x_e$), in which $s$ and $e$ are the start and ending frames of the span, and let $x_t$ be a frame in this span. The span boundary loss at time $t$ is computed as:

$$\mathcal{L}_{\text{SBO}}(z_t) = -\log\ p_{\text{span}}(z_t \mid h_s, h_e, p_{t-s}, q_{e-t+1}) \tag{2}$$

in which $h_s$ and $h_e$ are the output hidden states of the span boundaries $x_s$ and $x_e$, respectively, and $p$ and $q$ are the relative positional embeddings with respect to the left and right boundaries $x_s$ and $x_e$, respectively. The span prediction model $p_{\text{span}}$ is implemented as a 2-layer feedforward network followed by a softmax. The span boundary loss forces the model to predict the entire masked span without relying on individual tokens within it. The total loss is computed over all masked frames:

$$\mathcal{L}(\tilde{X}, M, Z) = \sum_{t \in M} (\mathcal{L}_{\text{MUM}}(z_t) + \mathcal{L}_{\text{SBO}}(z_t)) \tag{3}$$

# 4 PROSODYBERT FOR STYLE-CONTROLLABLE TTS

## 4.1 BACKGROUND: UTTS

We demonstrate the effectiveness of ProsodyBERT on UTTS (Lian et al., 2022b). UTTS is a framework for self-supervised multi-speaker TTS acoustic model pretraining that is combined with an alignment mapping model. The acoustic modeling pretraining does not rely on transcribed data. The alignment mapping model is trained on small amounts of paired text-audio data to map a phone-based forced alignment to a sequence of unsupervised units learned in pretraining. UTTS is developed from the perspective of disentangling speech representation learning. We choose UTTS because it is a self-supervised TTS framework that benefits from pretraining on a large amount of data, which aligns with ProsodyBERT's design. Also, our initial experiments indicate that UTTS synthesizes better quality speech than the trained models provided by Lee et al. (2022).

**Baseline UTTS Pretraining**  An overview of UTTS pretraining is shown in Figure 2. Speaker and alignment representation are disentangled via a conditional disentangled sequential variational autoencoder (C-DSVAE) (Lian et al., 2022a) during self-supervised training. Let $X$ be a training speech instance of length $n$ frames. Denoting model parameters as $\theta$, let $p_\theta$ and $q_\theta$ be the prior and posterior models, respectively. The training objective of C-DSVAE is a weighted sum of the following losses:

$$\mathcal{L}_{\text{KLD}_s} = \mathbb{E}_{p(X)}[\text{KLD}(q_\theta(z_{\text{speaker}} \mid X) \mid\mid N(0, I))] \tag{4}$$

$$\mathcal{L}_{\text{KLD}_c} = \mathbb{E}_{p(X)}[\text{KLD}(q_\theta(z_{\text{align}} \mid X) \mid\mid p_\theta(z_{\text{align}}))] \tag{5}$$

$$\mathcal{L}_{\text{reconstruct}} = \mathbb{E}_{p(X)}\mathbb{E}_{q_\theta(z_{\text{speaker}}, z_{\text{align}} \mid X)}[-\log(p_\theta(X \mid z_{\text{speaker}}, z_{\text{align}}))] \tag{6}$$

where $z_{\text{speaker}}$ is the learned utterance-level speaker representation (a vector embedding) and $z_{\text{align}} = [z_{a1}, z_{a2}, \cdots, z_{an}]$ is the learned phone alignment (a sequence of discrete unsupervised phonetic

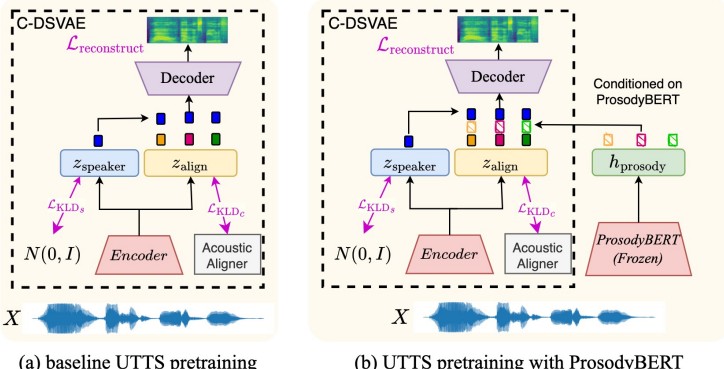

(a) baseline UTTS pretraining          (b) UTTS pretraining with ProsodyBERT

Figure 2: An overview of UTTS pretraining, which only requires raw speech. The loss terms during training are marked purple.

units $z_{ai}$). $\mathcal{L}_{\text{reconstruct}}$ is the reconstruction loss. KLD is the KL-divergence, and $\mathcal{L}_{\text{KLD}_s}$ and $\mathcal{L}_{\text{KLD}_c}$ encourage disentanglement between $z_{\text{speaker}}$ and $z_{\text{align}}$.

## 4.2 UTTS PRETRAINING WITH PROSODYBERT FEATURES

A limitation of the UTTS framework is that only speaker and word content (phone alignment) information are represented during training. There is no explicit model of prosody, making the synthesized demos less expressive and potentially unnatural. ProsodyBERT provides a solution. As shown in Figure 2 (b), during training, we condition the UTTS decoder on the prosody features extracted from the speech segment $X$. Formally, we replace Equation 6 with:

$$h_{\text{prosody}} = \text{ProsodyBERT}(X) \tag{7}$$
$$\mathcal{L}_{\text{reconstruct}} = \mathbb{E}_{p(X)} \mathbb{E}_{q_\theta(z_{\text{speaker}}, z_{\text{align}}|X)} [-\log(p_\theta(X \mid z_{\text{speaker}}, z_{\text{align}}, h_{\text{prosody}}))] \tag{8}$$

ProsodyBERT is designed to focus on prosody, disentangled from word content and speaker identity. By adding prosody embeddings ($h_{\text{prosody}}$) as a third source of reconstruction, the prosodic variations are accounted for. Given pretrained (frozen) ProsodyBERT embeddings, the C-DSVAE can focus on learning disentangled representations of the speaker ($z_{\text{speaker}}$) and content ($z_{\text{align}}$).

## 4.3 UTTS INFERENCE AND TRAINING WITH PROSODYBERT

**Inference** An overview of the UTTS inference pipeline with ProsodyBERT is in Figure 3 (a). Target speaker, text, and style ID are mapped into the learned representations of C-DSVAE to utilize the pretrained decoder for TTS. Lian et al. (2022b) describes the process of mapping a target speaker to $z_{\text{speaker}}$ and mapping a phone alignment to learned unsupervised alignment $z_{\text{align}}$. Here we focus on component models for predicting the prosody features and phone durations.

In TTS, prosodic structure is typically tied to words and/or syllables. Here, we use word-level prosody vector embeddings, but also explore phone-level vectors. In inference, the prosody features are first predicted given the text and the style, and then the duration of each phone is predicted given the style, the speaker, and the predicted prosody features. (As shown in Section 6, the inter-dependence of $F_0$, energy and duration make the prosody vectors useful for duration prediction.) After duration prediction, the prosody vector sequence is broadcast to the frame-level ($h_{\text{prosody}}$) according to the duration predictions. Similarly, the input phone sequence is broadcast to the frame level for input to the network that predicts the unsupervised acoustic unit sequence ($z_{\text{align}}$). The two prosody predictor configurations are shown in Figures 3 (b) (phone-level) and (c) (word-level). The architecture of the prosody and duration predictors follows the variance adaptor of Ren et al. (2021).

**Training** The UTTS decoder and acoustic unit mapping are trained as in Lian et al. (2022b), using the reference prosody vectors and learned unsupervised alignment. The prosody vector and duration predictors are separately trained. The reference phone- and word-level prosody vectors are the mean-pooled reference ProsodyBERT vectors using the respective (phone and word) forced alignment times. Unlike the baseline UTTS, our version with prosody conditioning learns the style vector during prosody prediction and not jointly with the decoder.

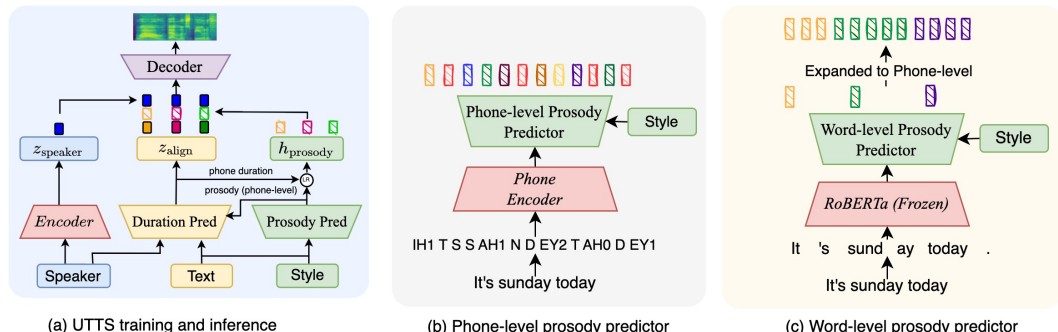

Figure 3: (a) UTTS for speech synthesis. The pretrained UTTS is paired with a duration predictor and a prosody predictor for speech synthesis. These predictors are trained on small amount of text-audio pairs; (b) and (c) show phone-level and word-level prosody predictors, respectively.

The configurations of the **prosody predictor** depends on whether the features are at the phone-level or word-level. The phone-level prosody predictor takes the phone embedding sequence and a style embedding as input, and it is jointly trained with the phone encoder and style embedding. The word-level predictor follows the design of Kenter et al. (2020). The input text is tokenized into WordPieces and passes through a RoBERTa encoder; the embedding corresponding to the first WordPiece of each word is used as the word representation. The word-level prosody predictor is trained jointly with the style vector using the frozen RoBERTa embeddings. Given $m$ phones or words (depending on the predictor), let the model output be $O = (o_1, o_2, \cdots, o_m)$ and the reference ProsodyBERT features be $H = (h_1, h_2, \cdots, h_m)$. The training loss is:

$$\mathcal{L}_{\text{prosody}} = \sum_{t=1}^{m} [\frac{1}{D}||o_t - h_t||_1 - \log \sigma(\cos(o_t, h_t))] \tag{9}$$

where $D$ is the dimension of hidden states, $\sigma$ is the sigmoid function, and cos is the cosine similarity. This loss is proposed in Chang et al. (2022), and we empirically find it yields good performance.

The **duration predictor** takes the phoneme sequence, the predicted prosody features (word-level features are expanded to phone-level via a lexicon), the style embedding, and the speaker embedding as inputs, and predicts the duration (number of frames) of each phoneme. Montreal Forced Alignment (MFA) (McAuliffe et al., 2017) is used to extract the target phoneme duration. The duration predictor architecture is the same as in Lian et al. (2022b) but with the added phone-level prosody input vector. The training loss is the mean squared error (MSE) between predicted duration and target duration (both transformed to the logarithmic domain).

## 5 EXPERIMENTAL SETUP

### 5.1 PROSODYBERT PRETRAINING

We use the full LibriTTS (Zen et al., 2019) audio for ProsodyBERT pretraining. LibriTTS contains 586 hours of audiobook data from 2,456 speakers. To generate the target labels, we run K-means clustering with 100 clusters using the `MiniBatchKMeans` algorithm in `scikit-learn` (Pedregosa et al., 2011). The mask span length is set to $l = 10$ for $20ms$ frames, and $l = 20$ for $11ms$ frames. Notice that the masks can overlap, so the real span length is variable. We choose this span length because the average word duration in human speech is about 200-300ms. For the sampling of mask starting point, we set the probability to be 65%, resulting in about $50\%$ of frames being masked. We reduce the model size and feature vector (to 32 dimensions) for computational efficiency. The ProsodyBERT architecture follows the distilBERT (Sanh et al., 2019) and has 21M parameters. We use Adam (Kingma & Ba, 2015) optimizer and linear learning rate schedule, with learning rate $1e-4$.

### 5.2 EXPERIMENTS ON TEXT-TO-SPEECH (TTS)

For text-to-speech experiments, two datasets are used: VCTK (Veaux et al., 2017) and DailyTalk (Lee et al., 2022). VCTK contains 44 hours of read speech from 109 speakers and does not contain much

prosody variation. DailyTalk contains 21.6 hours of spontaneous dialogue from 2 speakers and has rich prosody. We pool these two datasets so that the TTS system supports multiple speakers and two styles. VCTK reflects a reading speech style, and DailyTalk reflects a spontaneous speech style. The architecture of UTTS follows Lian et al. (2022b), in which the style and speaker embeddings are learned during training. To support zero-shot scenarios for speakers, we use the speaker embedding from pretrained ECAPA-TDNN (Desplanques et al., 2020; Zhang & Yu, 2022).

**Baselines.** We compare our TTS systems with two baselines. The first is the baseline UTTS, in which there is no explicit prosody component. For a fair comparison, we do not use extra audio for UTTS pretraining. The second is the official FastSpeech 2 checkpoint (Ren et al., 2021) trained on DailyTalk[2], the system that gets the highest MOS score in Lee et al. (2022), outperforming Tacotron 2 (Shen et al., 2018) and the authors' baseline. FastSpeech 2 contains a pitch predictor and a energy predictor that predicts quantized $F_0$ and energy, respectively. In contrast, our system is predicting ProsodyBERT features. FastSpeech 2 predicts phone durations from the phone sequence. Our UTTS with ProsodyBERT systems adds the predicted prosody features as inputs for duration prediction, which is shown to improve results.

### 5.3 Experiments on Emotion Recognition (ER)

We conduct the ER experiments on IEMOCAP (Busso et al., 2008) dataset, which contains 5 sessions. We follow the "leave-one-session-out" setting. In each round, one session is used for test and the others are used for training and validation. The evaluation metric is weighted accuracy, which is the accuracy of all utterances in the test session. We explore the use of ProsodyBERT features together with different pretrained acoustic encoders, including base and large versions of wav2vec 2.0 (Baevski et al., 2020), data2vec (Baevski et al., 2022), and HuBERT (Hsu et al., 2021). For all models, the base models have 12 layers with 768 dimensions each, and the large model has 24 layers with 1024 dimensions each. The acoustic encoder vectors are concatenated with 32-dimensional ProsodyBERT feature vectors.

There are two experiment settings. The first is the SUPERB (Yang et al., 2021) probing setting, in which the weights of the pretrained speech models are frozen. The SUPERB S3PRL probing model (an LSTM) takes the weighted average of each model layer (with supervision to learn weights) concatenated with the 32-dim ProsodyBERT features as the input. In the second setting, the concatenated frame-based sequence of acoustic and ProsodyBERT vectors is input to a conformer-based encoder-decoder module, and the model is jointly trained with an autoregressive setting to predict the emotion label and the corresponding transcription from the given utterance. To maximize the model performance, we experiment with HuBERT-large, since this model gives the state-of-the-art on IEMOCAP. We perform domain-adaptive ProsodyBERT pretraining on the training sessions of IEMOCAP, then finetune the whole model, including the pretrained acoustic model and our Prosody-BERT, via ESPnet[3] (Watanabe et al., 2018; Arora et al., 2022). As a contrastive experiment, we also use the raw pitch and energy prosody features instead of the ProsodyBERT vectors.

## 6 Results and Analysis

In this section, we first conduct speaker verification experiments to assess the extent to which the prosodyBERT learning strategy de-identifies the feature vectors, and then report results of TTS and emotion recognition experiments.

### 6.1 Speaker De-Identification

We assess the de-identification of pretrained ProsodyBERT features via a speaker verification task, where higher Equal Error Rate (EER) implies less speaker-related information in the features. The task is performed on the VCTK test set using the same code (including experiment configuration and random trials) as Lian et al. (2022a;b;c). We use a mean pooling of the frame-level prosody vectors as the representation of the speaker for an utterance. Two utterances are randomly drawn from

---

[2]`https://github.com/keonlee9420/DailyTalk`

[3]The hyper-parameters for training and decoding are the same as the default ESPnet recipe at `https://github.com/espnet/espnet/tree/master/egs2/iemocap/asr1`

the VCTK test set, and a cosine distance is used to determine whether the speakers are the same or different. The EER is computed from 36900 random trials. The ProsodyBERT features are compared to the full feature vector input to ProsodyBERT and the speaker-normalized prosody features used with K-means to derive the hidden prosody units. Table 1 shows that ProsodyBERT features have less speaker-identifying information than the full input acoustic-prosodic features (energy, pitch, and low-freq mel), as well as the speaker-normalized representations.

| Representation | EER |
|---|---|
| Energy + Pitch + Mel | 8.2% |
| Energy + Pitch + Mel (SN) | 20.4% |
| Energy + Pitch (SN) | 23.4% |
| **ProsodyBERT** | **35.3%** |

Table 1: Equal Error Rate (EER) of speaker verification on VCTK test set, using different prosody representations. SN means "speaker normalized." Higher EER implies a higher level of de-identification.

| | MOS($\uparrow$) | WER ($\downarrow$) | $L_{dur}$ ($\downarrow$) |
|---|---|---|---|
| Reference | $4.53 \pm 0.06$ | 5.6% | – |
| FastSpeech 2 | $3.37 \pm 0.09$ | 25.3% | – |
| Baseline UTTS | $3.50 \pm 0.09$ | 23.2% | 0.125 |
| Phone prosody | $3.63 \pm 0.08$ | 21.1% | 0.119 |
| **Word prosody** | **$3.81 \pm 0.09$** | **18.9%** | 0.117 |

Table 2: MOS scores (with 95% confidence intervals), ASR word error rate (WER), and duration validation loss ($L_{dur}$) for different TTS settings on DailyTalk.

## 6.2 STYLE-CONTROLLED TTS

**Quantitative Assessment**    Results of TTS experiments on the DailyTalk validation set are reported in Table 2 for the two configurations of ProsodyBERT in comparison to the two baselines (UTTS and FastSpeech 2) and the reference audio sample. Mean Opinion Score (MOS) tests use 20 subjects who are asked to evaluate the naturalness of each sample. Intelligibility is assessed in terms of ASR word error rate (WER) using a LibriSpeech ASR model.[4] For duration prediction, the validation loss (average MSE of log duration) is shown for the UTTS systems. Trends are consistent for all criteria, with higher MOS for the TTS systems using ProsodyBERT, as well as lower WER and duration loss. For all criteria, word-level prosody features give the best results, with a substantial improvement over FastSpeech 2. We hypothesize that this is due to the fact that word-level context is important for prosody prediction. The reduction in duration loss compared to the UTTS baseline shows that the prosody embeddings are implicitly representing duration effects.

**Example comparison of synthesized speech**    We compare the prosodic contours of the synthesized audio and the ground truth in detail on Figure 4. The $F_0$ plots show that both ProsodyBERT models capture the pitch accent on "go," but only the word-level prosody representation gives the final rise in $F_0$. The figures suggest that there are some issues with prediction of voicing with the ProsodyBERT outputs, but it is imperceptible and may simply be an $F_0$ tracking error. Energy contours also give a better match to the ground truth for the ProsodyBERT models. The spectrograms are given in Appendix C, showing that adding prosodic controls (both in FastSpeech and with ProsodyBERT) impacts the full spectral shape, not just $F_0$ and energy.

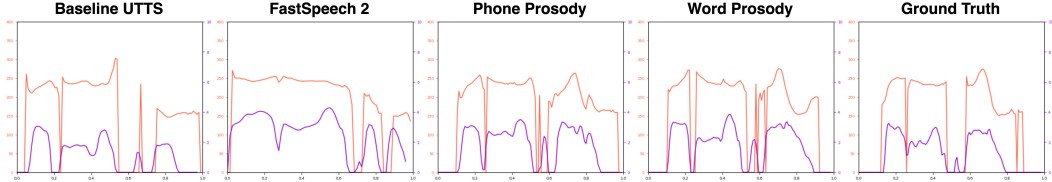

Figure 4: Comparison of the pitch and energy contours of synthesized speech "where do you want to go?" and the ground truth in DailyTalk (5_1_d1127). The orange line is $F_0$ and the puple line is $E$. The audio samples are available on the demo page.

**Expressiveness Control**    With the given training data, our system has two speaking styles: read (VCTK) and spontaneous (DailyTalk). However, more variation in expressiveness can be generated

---

[4]https://huggingface.co/espnet/simpleoier_librispeech_asr_train_asr_
conformer7_hubert_ll60k_large_raw_en_bpe5000_sp

using these style vectors, as shown in Figure 5. Specifically, we change the expressiveness by using prosody vectors that are a weighted sum of the word-level prosody features predicted for the two styles. The number in the figure titles are the weight of VCTK-style prosody features. As the weight becomes bigger, the synthesized speech has less variation in pitch and lower energy. Anecdotally, human evaluation suggests that the naturalness of synthesized speech is kept for all the weights.

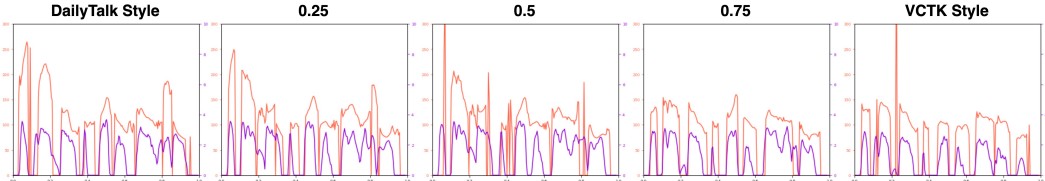

Figure 5: The pitch and energy contours conditioned on different level of expressiveness, controlled by weights of ProsodyBERT features of VCTK and DailyTalk styles. The text is "How can I solve the problem? I am angry."

## 6.3 EMOTION RECOGNITION ON IEMOCAP

| Model | Acc. | |
|---|---|---|
| | Original | +ProsodyBERT |
| wav2vec2-base | 63.4 | 64.2 |
| wav2vec2-large | 65.6 | 68.9 |
| data2vec-base | 66.3 | 70.4 |
| data2vec-large | 66.3 | 70.3 |
| HuBERT-base | 64.9 | 66.1 |
| HuBERT-large | 67.6 | 69.1 |

| Method | Acc. |
|---|---|
| WISE (Shen et al., 2020) | 66.5 |
| wave2vec2-PT (Pepino et al., 2021) | 67.2 |
| HuBERT-large (Gat et al., 2022) | 71.9 |
| HuBERT-large + TAP (Gat et al., 2022) | 74.2 |
| Ours | |
| HuBERT-large + Signal Prosody | 74.3 |
| HuBERT-large + ProsodyBERT | **75.8** |

Table 3: Results on IEMOCAP dataset in SU-PERB (Yang et al., 2021) probing setting.

Table 4: Results on IEMOCAP dataset in supervised setting. The "signal prosody" refers to ProsodyBERT inputs.

Table 3 shows our results on IEMOCAP in SUPERB probing setting, in which the pretrained models are all frozen during training. ProsodyBERT features are concatenated to the existing self-supervised acoustic model features as the input to the downstream models. In experiments with three different acoustic models, we find consistent improvement after adding prosody features, for both the base and large sizes. These results demonstrate that ProsodyBERT captures additional prosodic information that are complementary to all of these pretrained models.

Table 4 compares our results on IEMOCAP with those of recently reported systems in the fully supervised setting, in which all parameters are finetuned. Our method outperforms the current state-of-the-art (Gat et al., 2022). For analysis, we compare the ER accuracy of "HuBERT + ProsodyBERT" with "HuBERT + Signal Prosody" features, which includes $F_0$, energy, NCCF, deltas, and $< 500$ Hz log Mel. The relative performance gap shows the benefit of self-supervised learning.

## 7 CONCLUSION

We propose ProsodyBERT, a self-supervised method to learning prosody representations apart from speech content and speaker information. Our method does not rely on any transcripts or external models. This self-supervised framework allows ProsodyBERT to learn the full distribution of prosody variations via pretraining on large amount of data. We demonstrate the effectiveness of ProsodyBERT on a style-controllable TTS system, showing that the TTS model trained with ProsodyBERT features can generate natural and expressive speech, outperforming a model trained with energy and pitch. We also achieve a new state-of-the-art on IEMOCAP emotion recognition, showing that ProsodyBERT features are complementary to pretrained models like HuBERT, wav2vec2, and Data2vec. Future work may explore broader usage of ProsodyBERT. For example, ProsodyBERT may be combined with a large language model for spoken language understanding or translation tasks by augmenting text vectors with prosody embeddings.

## 8    REPRODUCIBILITY STATEMENT

The model details are given in Section 5. The source code for the model architecture, together with all the hyperparameters, are provided in the supplementary material. The trained model checkpoints will be released after the anonymity period, and they can be directly used with the provided source code. For the training procedure, we have accounted for the hardest issues in Appendix. The TTS demos uses in the MOS test are also provided in the supplementary material for reference.

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

# Appendices

## A  OVERVIEW OF DATA USED

| Dataset | Domain | Hours | Total speakers | Usage |
|---|---|---|---|---|
| LibriTTS (Zen et al., 2019) | Audiobook | 586 | 2,456 | ProsodyBERT pretraining |
| VCTK (Veaux et al., 2017) | Reading | 44 | 109 | Text-to-speech |
| DailyTalk (Lee et al., 2022) | Conversation | 21.6 | 2 | Text-to-speech |
| IEMOCAP (Busso et al., 2008) | Conversation | 12.5 | 10 | Emotion Recognition |

Table 5: Summary of the datasets we used.

An overview of the data we use are shown on Table 5.

## B  MORE ABOUT DURATION PREDICTION

| Inputs | Val loss (predicted prosody) | Val loss (oracle prosody) |
|---|---|---|
| Phones | 0.125 | - |
| Phones + Phone Prosody | 0.119 | 0.010 |
| Phones + Word Prosody | **0.117** | 0.077 |

Table 6: The validation loss of duration prediction on DailyTalk. The loss is computed by MSE on log duration (number of frames of each phone, with frame length about 11ms).

Table 6 shows the duration prediction performance for the baseline UTTS model compared to prediction using prosody vectors, both for automatically predicted and oracle prosody. The results show that the prosody vectors reduce prediction error, which demonstrates that duration effects of prosody are implicitly modeled by mean-pooling the sequence of frame-level vectors.

## C  SPECTROGRAMS OF SAMPLES

We include the spectrograms of the audio samples presented in Section 6 in Figure 6 and Figure 7 for reference. Figure 6 shows that besides the energy and pitch contours, prosodic controls also impact the full spectrogram for both UTTS and FastSpeech 2.

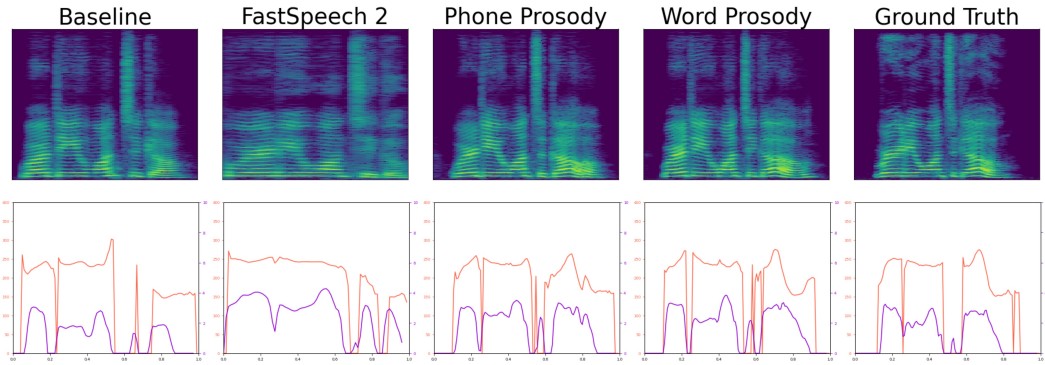

Figure 6:  The log Mel spectrograms of the audio samples in Figure 3.

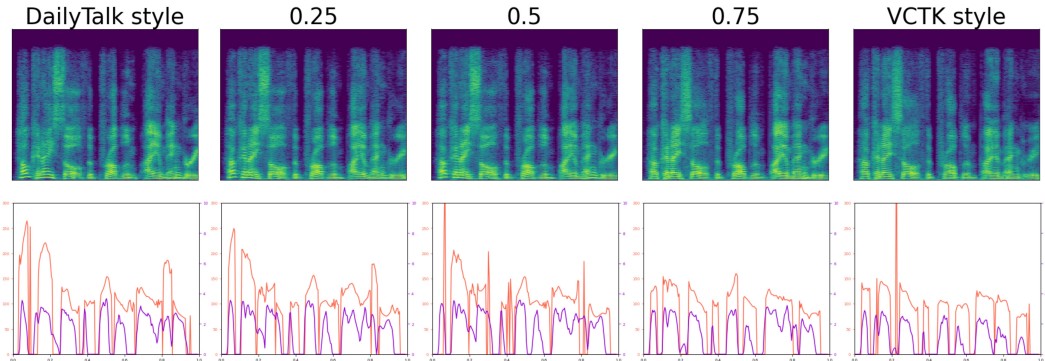

Figure 7: The log Mel spectrograms of the audio samples in Figure 5.

# D IMPLEMENTATION DETAILS

ProsodyBERT has two configurations. For emotion recognition, following Hsu et al. (2021), the input audio sampling rate is 16kHz, and the frameshift is 20ms. For TTS, following Ren et al. (2021), the input sampling rate is 22.05kHz, and the hop length is 256 (frame shift $\sim$ 11ms). For both the ER and TTS versions, we train the model for 10 iterations on 8 GPUs, with batch size 8. Training takes about 6 hours for ER and 10 hours for TTS (due to the shorter frame shift).

## D.1 ARCHITECTURE

The ProsodyBERT architecture follows the distilBERT architecture (Sanh et al., 2019), with the exception that the positional embeddings are replaced with the convolutional positional embedding as in (Baevski et al., 2020) to support long inputs. The encoder has 6 transformer layers, with hidden dimension 512, 8 heads, feedforward dimension 2048. The final projection layer has dimension 32. The model has 21M parameters.

## D.2 K-MEANS ON ACOUSTIC CORRELATES

We train a K-means clustering model on frame-level prosody features (NCCF, $\log(F_0)$, $\Delta \log(F_0)$, $E$). Z-score speaker normalization is performed on $\log(F_0)$ and $E$, and $\Delta \log(F_0)$ is computed on the normalized $\log(F_0)$. NCCF is already normalized, so no further normalization is needed. By such, we disentangle speaker information. Each feature is a scalar, so clustering is done on 4-dim vectors. We have not added any weightings on each dimension because the features have already been normalized before clustering.

## D.3 SPAN MASKING

We adopt the same masking mechanism as SpanBERT (Joshi et al., 2020), wav2vec 2.0 (Baevski et al., 2020), and HuBERT (Hsu et al., 2021). Let the input utterance be $X = (x_1, x_2, \cdots, x_n)$, we first randomly select a subset of $Y \subseteq X$ such that $|Y| = m|X|$. $m$ is a given hyperparameter, and it is set to $65\%$ in our experiments. The frames in $Y$ are the starting point of the masked spans, and spans of $l$ frames are masked. Notice that the spans can overlap, so the length of the spans is not fixed.

## D.4 ALIGNMENTS OF WORDPIECE, PHONEMES, AND WORDS

There is no direct mapping between WordPieces and Phones. However, the RoBERTa encoder in word-level prosody predictor only takes WordPiece as inputs, and the UTTS system takes frame-level inputs. To deal with this mapping issue, we first map words to WordPieces (represent each word by its first WordPiece, and ignore the other WordPieces), and then broadcast this word-level feature to its corresponding phones via a lexicon.

