# OpenReview forum: "ProsodyBERT: Self-Supervised Prosody Representation for Style-Controllable TTS"
_ICLR.cc/2023/Conference — Submitted to ICLR 2023_

### Official Review · Reviewer_CWFe · 2022-10-22

**Confidence:** 4
**Correctness:** 3
**Technical Novelty And Significance:** 2
**Empirical Novelty And Significance:** 3
**Recommendation:** 5

**Clarity, Quality, Novelty And Reproducibility:**

Overall, the technical portion of this work was fairly easy to follow. I generally trust that improvements have been made. I can certainly hear some improvements in the provided audio samples.

I don't think the novelty of this work is very high. Algorithmically, vector quantization (k-means) has been explored and used for resynthesis in [1]. This work takes it a bit further by incorporating additional features such as log F0, energy, and NCCF.

[1] Polyak et al.  Resynthesis from Discrete Disentangled Self-Supervised Representations. Interspeech 2021


**Strength And Weaknesses:**

Strengths:
- Fairly simple setup to achieve explicit incorporation of prosodic features
- Demonstrated applicability in several downstream tasks

Weaknesses:
- While I have no doubt their setup works in practice, I do have some concerns regarding the K-means clustering.
  - Not all frames contain valid F0 values. It's not clear how this is being handled, but constants and interpolated values will introduce devations from the true data distribution
  - It's also not clear how to best combine these individual attributes. Scaling individual features can have drastic effects on which one dominates the clustering task.
- I don't think there's an ablation on the SpanBert objective
- Error bars missing in Table 1. Furthermore, MOS is not a good metric for comparing fine grained differences between various models compared to performing pairwise preference testing.
- While there is no known good way to truly measure the quality of synthesized F0 and energy, I would be very careful about making claims backed up by a single qualitative comparison in Figure 4.
- I'm not sure I follow the logic behind the claim that wave2vec, hubert, and waveLM filter out long-range prosodic information.
- It's not fully clear how necessary the large transformer architecture with the self supervised objectives matter at all for some of the downstream tasks. I suppose an even simpler baseline would've been to individually quantize each prosodic feature into K bins using K-means (separately per feature). Then for each downstream task, pair each 1-hot encoding with its own embedding matrix to map to a lower dimensional representation. Alternatively, perhaps for the emotion recognition task, one could even just directly condition on the raw values like in Mellotron (Valle 2020), appending the user-normalized raw values with HuBERT features.

Questions for clarity:
  - Can you clarify how EER is computed?

**Summary Of The Paper:**

The following work proposes a self-supervised representation for learning. Unlike prior methods, the proposed framework relies only on raw audio. Feature vectors are extracted based on a combination of NCCF, log F0, delta log F0, and average energy per frame. The resulting N-D feature vector is k-means clustered to achieve discrete codes. Self-supervised training objectives from HuBERT and SPANBert are then used to learn the final representation. The resulting features are then applied to downstream unsupervised TTS and emotion recognition.

**Summary Of The Review:**

Overall, easy to read paper with some promising results. However, I think the novelty is somewhat low and the complexity of the solution (large transformers with self supervised learning objectives) might not be necessary for some of the downstream tasks demonstrated in this work. However, I'm certainly happy to change my review if the authors can demonstrate otherwise.

---

> ### Author Response · Authors · 2022-11-17
> **Thanks for your review**
>
> Thanks for your detailed and helpful comments! We have revised our draft to address your concerns:
>
> **[About Novelty. Is this another work about vector quantization?]**
>
> We really appreciate this comment, which enabled us to see this possible confusion. As noted in the summary comments above, this work is not simply doing vector quantization. K-means just provides the inductive bias for self-supervised learning to derive vector embeddings. The figures and the methods section have been revised to make this clearer.
> As we have shown, the self-supervised features have extra benefits in de-identification (see Section 6.1) and implicitly learning duration/rhythm information (see section 6.2 and Appendix B). Subjective TTS evaluation suggests that it surpasses the baseline UTTS and the simple energy/pitch quantization used in Fastspeech 2. Polyak et al. (2021) train a VQ-VAE on F0 for quantization, which is much more limited compared to what our features present. We also update the Intro and Related work sections to discuss the novelty of our work compared to prior work.
>
> **[Is it necessary to use SSL for the downstream tasks?]**
>
> FastSpeech 2 models prosody in a way similar to what you mentioned, by quantizing prosody and energy separately. We have added a discussion about baselines in Section 5.2. FastSpeech 2, despite working well on datasets like LJSpeech and LibriTTS, does not work well on DailyTalk because of the new dataset’s rich prosody (see the MOS scores and the demo page). What we present in this paper is the system that we find working the best on DailyTalk. We believe that the self-supervised ProsodyBERT feature is playing a key role in the improvement of prosody modeling.
>
> Also, we would like to point out that using ProsodyBERT features in TTS system brings almost no extra inference cost for speech synthesis. The transformer is only used for training data generation (32-dim prosody features), and the TTS system does not need the ProsodyBERT encoder for inference.
>
> For the ER task, we believe the “signal prosody” accounts for the raw values you mentioned, and the relative performance gap shows the benefit of SSL. Besides, thank you for bringing up Mellotron! It is an important relevant work and we add it to the Introduction and Related Work sections in the revised paper.
>
> **[About K-means clustering]**
>
> Following FastSpeech 2 and Kaldi, we use interpolated F0 values. So, our self-supervised learning objective is learning a pitch contour. For clustering, we have performed speaker-level z-score normalization on the 4 acoustic-prosodic features used for K-means. z-score normalization is effectively a weighting. We have updated Section 3 to clarify these concerns.
>
> **[About writing]**
>
> We have revised the paper to address your concerns and rewrote/deleted the claims that are not rigorous. We also added error bars to the MOS table.
>
> **[About EER]**
>
> We add a more detailed description of EER experiments in Section 6.1.
>
> **[Summary]**
>
> Again, thanks for your careful and valuable comments! They point out the aspects that we haven’t explained well in the original draft and help strengthen our paper during revision. We hope that our comments and revisions have addressed your concerns!
>
> > References
>
> > [1] Adam Polyak, et al. "Speech resynthesis from discrete disentangled self-supervised representations." Interspeech 2021.
>
> > [2]  Rafael Valle, et al. “Mellotron: Multispeaker expressive voice synthesis by conditioning on rhythm, pitch and global style tokens.” ICASSP 2020.

---

### Official Review · Reviewer_PaY1 · 2022-10-24

**Confidence:** 4
**Correctness:** 4
**Technical Novelty And Significance:** 4
**Empirical Novelty And Significance:** 4
**Recommendation:** 8

**Clarity, Quality, Novelty And Reproducibility:**

The paper is clear even though the complexity of the proposed approach is high. The novelty is clear, both in terms of the methods and performance improvement in unsupervised TTS. The proposed approach is not easily reproducible due to the inherent complexity of the proposed approach, the multiple stages of the pipeline, and the complexity of the task under study. The authors included a statement with a pledge to share checkpoint models and code with hyperparameters values and source code for training. Either way, the stage related to phone and word alignment could be better explained.

**Strength And Weaknesses:**

The paper exploits recent ideas in speech processing and text-to-speech technologies, to propose a new and promising idea to incorporate prosodic information into speech synthesis/generation tasks. The proposed approach is theoretical well supported, and the authors provided experiments and results with enough empirical evidence of the improvements achieved by their approach. Nice work.

**Summary Of The Paper:**

The paper proposes a BERT-based architecture for extracting prosody embeddings from speech. The pre-training of the model is performed following a self-supervised strategy similar to that of HuBERT but based on prosody features to get the cluster labels. The performance of the proposed approach is evaluated in an unsupervised Text-To-Speech Task proposed in previously published work.

**Summary Of The Review:**

The paper proposed a BERT-based architecture and a self-supervised training strategy for prosodic-related embedding representation extraction from speech signals. The prosodic learned features are used in an unsupervised TTS task yielding a real improvement in terms of Mean Opinion Score and an ASR Intelligibility metric (Word Error Rate). The proposed approach is built on top of recent advances in speech processing and unsupervised TTS technologies and proposed new ideas to incorporate prosodic features to enhance the natural and expressive of synthetic speech.

This is a great work, congratulations to the authors.

---

> ### Author Response · Authors · 2022-11-17
> **Thanks for your review**
>
> Thanks for your appreciation of our work and careful review!
>
> **[Phone and word alignment could be better explained]**
>
> We have revised Section 4 to make the discussion about the phone and word alignment part clearer.
>
> **[Summary]**
>
> We are honored that you believe that this is great work, our approach is theoretically well-supported, and the experiments provide enough empirical evidence. We are excited that our new prosody feature brings a big improvement to the naturalness and expressiveness of TTS, and believe that it will be inspiring for future works. Again, thanks for your appreciation on the value of our work!

---

### Official Review · Reviewer_J32w · 2022-10-26

**Confidence:** 5
**Correctness:** 2
**Technical Novelty And Significance:** 2
**Empirical Novelty And Significance:** Not applicable
**Recommendation:** 3

**Clarity, Quality, Novelty And Reproducibility:**

Clarity:
 - Writing style needs improvement. For example, the correct use of `\citep{}` and `\citet{}`
 - It's not clear how "6.1.2 EXPRESSIVENESS CONTROL" is conducted.

Quality:
 - I have concerns on the technical correctness. For examples:
   - This work claims to learn prosody representation. However, key prosody components, e.g. pace / rhythm, are not concerned in the proposed model.
   - The TTS experiment used FastSpeech 2 as one of the baseline, and claimed that used "the official FastSpeech 2 checkpoint". However, there is no such "official FastSpeech 2 checkpoint" ever released.
   - Even if the authors used a non-official third-party FastSpeech 2 checkpoint, it's unlikely that the checkpoint was trained on the same dataset as used in this work (DailyTalk), which is not a popular TTS dataset. As such, the comparison in Table 1 is not fair at all.
   - "The most commonly used prosody representations include acoustic attributes like fundamental frequency (F0), energy, and duration." Please provide references.
   - Fig 2 doesn't make well sense -- it looks like the TTS prediction only depends on z_speaker, z_align, z_prosody but nothing else. How about content?
 - I also have other concerns on the experiment setup.
   - For the TTS experiment -- are the improvement from using extra features extracted by another model pre-trained with extra data, or from using extra features extracted by ProsodyBERT pre-trained with extra data? A comparison between UTTS + HuBERT and UTTS + ProsodyBERT would be required to show the improvement are from ProsodyBERT.

Novelty:
 - Limited. The proposed model is similar to HuBERT / SpanBERT, except for adding more input features (f0, energy, NCCF).

**Strength And Weaknesses:**

Strength:
 - The idea of this work is sound
 - The experimental results looks interesting

Weaknesses
 - The novelty is limited.
 - Technical correctness issues, as explained in the next section.

**Summary Of The Paper:**

This work explores utilizing self-supervised pretraining on speech data for improving prosody modeling in TTS and emotion recognization tasks. The proposed model, ProsodyBERT, is similar to HuBERT / SpanBERT, but with more input features added (f0, energy, NCCF). Experiments are conducted on two tasks: TTS (on DailyTalk + VCTK datasets) and emotion recognition (on IEMOCAP dataset).

**Summary Of The Review:**

This paper tackle on an important problem (self-supervised prosody modeling). However, the contribution from the works is limited, and there are issues on the technical correctness.

---

> ### Author Response · Authors · 2022-11-17
> **Thanks for your review**
>
> Thanks for your feedback, and for acknowledging that we are tackling an important topic. We understand that your concerns are mainly related to the paper’s novelty and technical correctness, and hope our response resolves your concerns.
>
> Before addressing your concerns in detail, we would like to clarify the focus and contributions of this paper first.
>
> **[Novelty]**
>
> On the representation learning side, our main contributions are:
> 1. Proposing an effective SSL method that focuses on **prosody-related variations**, rather than acoustic-phonetic information (which HuBERT focuses on).
> 2. **Disentangling** prosody from the speaker and content information without relying on transcripts.
> 3. Evaluations on TTS and SOTA performance on IEMOCAP show that ProsodyBERT amplifies prosodic information and normalizes speaker and content variabilities.
>
> On the TTS side, our main contributions are:
> 1. We show that predicting prosody features at the **word level** achieves better naturalness compared with the **phoneme level**. This finding agrees with the theory that prosody is related to higher levels of context, and could be beneficial for future TTS works.
> 2. We **improve the duration model** adopted widely after FastSpeech 2 by adding prosody representation to the input of the duration prediction model, and demonstrate that it improves the duration/rhythm of synthesized speech.
> 3. With all these new designs, we are able to generate speech with largely improved prosody naturalness and diversity.
> 4. With the ProsodyBERT and UTTS framework, we illustrate prosody control ability to a certain extent. Due to the dataset limitation, we can only demonstrate two styles in this paper. Nevertheless, our prosody representation interpolation experiments show great potential for using ProsodyBERT to achieve **fine-grained controllable TTS**. For example, our system may be used to control dialect density and emotion intensity.
>
>
> **[About FastSpeech 2 Baseline]**
>
> There are official baselines for TTS trained on DailyTalk, provided by the authors of DailyTalk in the official repo. The link is here: [https://github.com/keonlee9420/DailyTalk](https://github.com/keonlee9420/DailyTalk). We also add a paragraph of baselines in Section 5.2. We are comparing with the checkpoint that achieves the highest MOS score in the DailyTalk paper.
>
> **[About TTS. How is content represented?]**
>
> This is a misunderstanding that may lead to questions about the correctness of our work. Content is carried by the phone alignment (which is a sequence of phones repeated according to phone durations). It is represented by z_align in the paper. Please refer to Section 4 for more details.
>
>
> **[Pace/rhythm component of prosody]**
>
> We add a note in the intro that duration aspects of prosody are implicitly represented in the temporal sequence, and the word/phone-level prosody features are used to predict the duration. (Rhythm is a combination of duration and energy patterns, so we claim that it is captured with our features.) We also expand the results table in 6.2 to include duration loss, demonstrating that the prosody vectors improve duration prediction. Appendix B provides more quantitative results on duration prediction. In particular, duration prediction loss is reduced from 0.125 to 0.010 when adding oracle prosody features to the inputs.
>
> **[Clarity: writing style, expressiveness control]**
>
> We have revised the paper to fix these concerns and some other issues in the bibliography. Regarding the expressiveness control experiments, we have clarified that the observations are based on our assessments of a small number of examples, and we provide additional examples in the demo for readers to draw their own conclusions.
>
> **[UTTS + HuBERT baseline]**
>
> HuBERT features do not fit UTTS architecture well, because UTTS is designed to disentangle speaker, phonetic, and prosody information. If we are using HuBERT in the prosody branch, then the phonetic branch of UTTS will be poorly trained because HuBERT contains too much phonetic information. Meanwhile, the key contribution of ProsodyBERT is disentangling prosody from other factors. So, we believe that UTTS+HuBERT will work badly, and this is not a fair comparison.
>
> **[Summary]**
>
> Again, we appreciate your valuable comments and believe that there are some misunderstandings during the process. We have addressed them carefully in this rebuttal, and hope that it helps resolve your concerns!

---

> > ### Comment · Reviewer_J32w · 2022-11-29
> > **Reply**
> >
> > Thanks for the responses from the authors. They do address some of my concerns on: 1) If the TTS baseline is trained on the same dataset (DailyTalk); 2) How the TTS model is conditioned on content.
> >
> > My concerns on novelty and effectiveness remain:
> >  - Novelty is limited. The proposed model is similar to HuBERT / SpanBERT, except for adding more input features (f0, energy, NCCF).
> >  - Effectiveness compared to existing methods is unclear. The TTS experiment set up in this paper is quite specialized (to a very recent UTTS model). Can it help other TTS models (e.g. FastSpeech)? Does it perform better than existing methods? Are the improvement primarily from the fact of using extra features / extra data / extra model capacity, or from the specific pre-training method proposed by this work? Regarding comparison to UTTS + HuBERT -- while HuBERT output is high dimensional indeed, but what if simply projecting its output to a low dimensional space, or using an attention (https://arxiv.org/abs/1811.02122) or a fine grained VAE (https://arxiv.org/abs/2002.03788) to align phone/word with HuBERT outputs?
> >
> > Aside for the major concerns, authors' statement on "the official FastSpeech 2 checkpoint (Ren et al., 2021)" is still misleading, as it's not released by Ren et al.

---

> > > ### Author Response · Authors · 2022-11-29
> > > **Reply from authors (Relation to HuBERT)**
> > >
> > > Hi Reviewer J32w,
> > >
> > > Thanks for your response! It's good that some of your concerns are addressed. For your other concerns, we believe there are still some misunderstandings:
> > >
> > > **[Novelty]**
> > >
> > > First, we want to emphasize that **ProsodyBERT feature learning is just 1/3 of this paper**.
> > > We have also done a lot of innovative work on how to combine this SSL framework with TTS systems (which is not explored in HuBERT), analysis of the learned prosody representation, and emotion recognition. Our experiments and demos have shown stunning improvement in use of prosody over prior systems.
> > >
> > > Second, we are not simply adding prosodic features, we are also **removing phonetic and speaker information**. This **disentangling** part is one of our major technical contributions; it is critical for ProsodyBERT to provide complementary information to the phonetic information represented by HuBERT. To achieve this goal, compared with HuBERT/SpanBERT, our innovations are:
> > > 1. ProsodyBERT's pseudo-labels are K-means on the speaker-normalized acoustic-prosodic features, **NOT MFCC**. HuBERT uses MFCC, which can be viewed as a pseudo-label for local phonetic labels. Such a design makes **HuBERT focus on phonetic information** and thus is good for speech recognition. In contrast, the design goal of ProsodyBERT is to **FOCUS on prosody** and **REMOVE phonetic information and speaker information**.
> > > 2. To remove phonetic information, we also **remove high-frequency information** from the inputs. To achieve this goal, we **changed the model architecture** and replaced the CNN encoders.
> > > 3. The span boundary loss is added to address the suprasegmental nature of prosody and capture the interdependence of F0, energy with duration. Experiments on duration prediction in Section 6.2 and Appendix B.
> > > 4. ProsodyBERT focuses on prosody while HuBERT focuses on phones. These two models are learning two features that are different both in theory and practice. The input, output, model architecture, and loss function are all changed and extensive tuning has been done to make this work. Many design decisions are made based on theories of prosody and phonology. We actually have rewritten the whole training pipeline (which will be released) to make this work. It is not a trivial "add features to the input and use old code base" work.
> > > 5. Moreover, prosody has long been a hard issue to deal with because it is difficult to disentangle with other factors. Besides the technical contributions, we believe **the idea of using SSL to learn disentangled prosody features alone has sufficient novelty**. And based on our experiments, ProsodyBERT is very ease-to-use and effective for both TTS and ER. We believe it will benefit many follow-up studies.
> > >
> > >
> > > **[About UTTS + HuBERT]**
> > >
> > >
> > > The major issue is not the number of dimensions, which can be addressed by the methods you mentioned.
> > > The problem is that **HuBERT features are NOT disentangled with speaker information and phonetic information**, which will cause trouble in the controllable TTS training (the three branches of C-DSVAE should be mutually exclusive).
> > >
> > >
> > >
> > > Again, thanks for your reply! We are happy to discuss more if you have further questions.

---

> > > > ### Author Response · Authors · 2022-12-03
> > > > **Reply from Authors (TTS Issues)**
> > > >
> > > > **[On TTS comparison]**
> > > >
> > > >
> > > > We agree that it would be useful to assess ProsodyBERT in other TTS systems, and our plan is to explore this in future work with an opensource model such as FastSpeech. However, we argue that it is at least as important to demonstrate advances in a state-of-the-art system, and our work shows that the UTTS baseline outperforms FastSpeech. Further, it is important to make controlled comparisons, as is the case for the experiments using the baseline UTTS and the UTTS+ProsodyBERT. They share the same model architecture (except 32 additional dimensions for the transformer input) and the same training corpus. The major difference is the 32-dim ProsodyBERT features used during training. The improvements due to ProsodyBERT are substantial and clearly demonstrate the effectiveness of ProsodyBERT features.
> > > > We add FastSpeech 2 to give readers a sense of how well the most popular expressive TTS models work on this dataset.
> > > >
> > > >
> > > > **[Transferable to other TTS systems]**
> > > >
> > > >
> > > > Our system shares many components with other TTS systems (e.g. duration predictor, variance predictor, vocoder, decoder architecture...) So, we believe that the improvements and analyses we made on UTTS are transferable.
> > > > For example, the improved duration predictor and the word-level prosody predictor can be adopted by other systems.
> > > > Most current successful TTS systems are really complex and have numerous components. Since we also wanted to demonstrate utility of ProsodyBERT in emotion recognition, the extensive effort of experimenting with multiple TTS systems is beyond the scope of this paper.  We hope future work can explore more in this direction.
> > > >
> > > >
> > > > **[Checkpoint of FastSpeech 2]**
> > > >
> > > >
> > > > Our revised paper says
> > > > > The second is the official FastSpeech 2 checkpoint (Ren et al., 2021) trained on
> > > > DailyTalk, the system that gets the highest MOS score in Lee et al. (2022)
> > > >
> > > > We also have added the link to the checkpoint. Here the word "official" refers to the dataset authors (Lee et al., 2022), not the authors of FastSpeech 2  (Ren et al., 2021).
> > > > We are happy to rewrite this sentence if you believe this is misleading.
> > > >
> > > >
> > > > Again, we appreciate the points raised here. We do hope to demonstrate the benefit of ProsodyBERT more broadly to other TTS systems (as well as in speech understanding tasks) in future work.

---

### Official Review · Reviewer_925W · 2022-11-01

**Confidence:** 4
**Correctness:** 3
**Technical Novelty And Significance:** 3
**Empirical Novelty And Significance:** 3
**Recommendation:** 5

**Clarity, Quality, Novelty And Reproducibility:**

Clarity

The paper is overall well-written. As it builds on a lot of prior work, at times it becomes understandably hard to be self-contained, and some components of the modeling are only mentioned in passing. For example, what is “style”, in addition to speaker? What is z_align? At times, it took a while to figure out what was proposed in prior work and what was new. For Figure 2, would it be possible to visually distinguish which modules are new and which modules and pipeline are building on prior work. The current dotted-lined box in Figure 2(a) is helpful. Is Figure 2(c) a zoomed in illustration of the “shared predictor” module in Figure 2(b)?

Clarification questions on the K-means procedure: What is the dimensionality of the extracted features, and how are they weighted when performing K-means? Was there a need to explore different weightings or dimensionality reduction? How are the interdependencies between the different features accounted for? What is the actual dimensionality that the K-means was performed in?

Quality

The proposed approach is well-thought-out. The experiments include both quantitative, qualitative and subjective evaluations. Figures such as Figure 4 helps the reader understand the differences in what the approaches are able to capture, which is very helpful. More discussion on comparison to prior work and samples could strengthen the quality of the work.

Novelty

In contrast to prior work that used mostly pitch and energy when considering prosody, this paper proposes to use more expressive domain-informed features to better capture prosody, and to model it using a HuBERT-like approach, making it possible to learn general prosody from large amounts of (unpaired) speech audio.

Reproducibility

The description of ProsodyBERT is detailed, and the TTS pipeline builds on prior published work. The code for ProsodyBERT is included in the supplementary material, and it was mentioned in the paper that the checkpoint will be released at a later point.


**Strength And Weaknesses:**

Strengths

This paper provides a targeted and effective solution for modeling prosody, which is crucial to synthesizing expressive and natural speech. The overall design of the modeling, training and inference pipeline is well-thought-out, leveraging strong prior work as building blocks, such as HuBERT (Hsu et al, 2021) and SpanBERT (Joshi et al, 2020) for prosody representation learning and UTTS (Lian et al, 2022a,b) as the unsupervised text-to-speech synthesis, and demonstrates how ProsodyBERT representations can be added to strength prosody generation. The provided synthesis examples are compelling, and the improvement on the emotion recognition task shows that the choice of features and learned representations are useful.

Weaknesses

Since the main contribution of the paper is on learning prosody representations and predicting prosodic features for synthesis, it could be helpful to provide more grounding on what prosody is, and how it motivates subsequent design choices in modeling. Informally, prosody is both a general phenomenon, could also be a person’s signature way of speaking. What is the dependency structure between style, speaker, prosody, duration? How does the paper choose its current ordering of conditioning and predicting them? In the appendix, zero-shot speaker synthesis was mentioned as a motivating factor for prosody features to be not dependent on speaker. Perhaps this is a design choice inherited from prior work that focuses on the zero-shot setup that is also well-suited for the current setting?

For comparison to prior work, it could be helpful to have a paragraph discussing how the baselines are chosen, what other models are considered but may not be as relevant, or practically not feasible to compare to. In what ways it may or may not be an apples-to-apples comparison. For example, what are the model sizes involved, which may be a result of the different design objectives of the prior models?

Some examples seem to be missing from the demo page, for example “emotion control” examples, and there’s only one example for “expressiveness control”. For the “expressive TTS” examples, would it be informative to include a few examples for each text to show the range of expressiveness that is possible. It seems all the text sentences are from the original dataset. How does the approach perform on unseen text?


**Summary Of The Paper:**

This paper introduces ProsodyBERT for learning prosody representations to strengthen the prosodic aspects in unsupervised text-to-speech synthesis (TTS). Similar to HuBERT (Hsu et al, 2021), ProsodyBERT is a masked encoder model trained to predict K-means cluster labels, except in this case the K-means is performed on domain-informed features (such as pitch, energy, and dynamics, etc) to capture the long-term temporal structure of prosody. These learned hidden features then become targets for training a prosody predictor, and then downstream a duration predictor. To evaluate the effectiveness of ProsodyBERT, the paper demonstrates expressive TTS, interpolation between speaker’s prosodic styles, and how ProsodyBERT features improve emotion recognition in speech.


**Summary Of The Review:**

The paper brings to attention the importance of prosody, and provides an effective approach to modeling it and adding it to unsupervised TTS systems. The approach is sound, and the results sound compelling. More discussion on the evaluation and comparison to prior work could help position this work and its contributions. A more complete demo page would also strengthen the paper’s contributions.

---

> ### Author Response · Authors · 2022-11-17
> **Thanks for your review**
>
> Thank you for your kind feedback and valuable comments! We have revised our draft and the demo page to address your concerns:
>
> **[More about prosody. How does it motivate subsequent design choices?]**
>
> Following your suggestions, we have expanded the definition of prosody in the introduction and added a paragraph motivating the disentangling of different factors, including better generalization as well as zero-shot speaker synthesis. The choice of making prosody features speaker-independent is also motivated by the benefit for privacy-preserving processing.
> The dependency structure between style, speaker, text, prosody, and duration is now more clearly described in Section 4.3. We also add quantitative results on duration prediction and more discussions on the example contours in Section 6 to demonstrate the utility of our design.
>
> For your reference, here is a summary of the novelties of our TTS design. The updated draft has addressed these points:
> 1. We use **ProsodyBERT features as supervision signals**, rather than quantized energy and pitch. The improvement is shown by the MOS scores and the example contours in Section 6.
> 2. We show that predicting features at the **word level** achieves better naturalness compared with the **phoneme level**. This finding agrees with the theory that prosody is related to higher levels of context, and could be beneficial for future TTS works.
> 3. We improve the widely adopted FastSpeech 2 duration model by **adding prosody representation to the input of the duration prediction model**, and demonstrate that it improves the duration/rhythm of synthesized speech in Section 6 Table 2 and Appendix B.
> 4. Our prosody feature interpolation experiments show great potential for using ProsodyBERT to achieve **fine-grained controllable TTS**. For example, our system may be used to control dialect density and emotion intensity.
>
>
> **[Comparison with baselines]**
>
> We have added a baseline description paragraph in Section 5.2. Specifically, we give a detailed comparison of our work with FastSpeech 2.
>
> **[Demo page]**
>
> We add more samples on unseen texts and expressiveness control on the demo page.
>
> **[Clarity about TTS]**
>
> Following your suggestions, we have made a major revision to Section 4 (about TTS). We added more discussion on the style and z_align, changed Figure 2 as you suggested, and replotted the “shared predictor” in Figure 3(a) to make it clearer.
>
> **[Clarity about K-means]**
>
> We have revised the feature extraction part in Section 3 to more clearly specify that the 4 acoustic-prosodic features (with speaker-level z-score normalization) form the 4-dim vectors used in K-means clustering. The z-score normalization is effectively a weighting, and no dimensionality reduction is needed since the dimension is already small.  The interdependence of the full set of acoustic features (4 prosodic + 20 low-frequency Mel Spectral bins) and duration is implicitly represented via self-supervised learning.
>
> **[Suggestions on Quality]**
>
> We have expanded the discussion of prior work and the example contours, and added more audio samples on the demo page.
>
> **[Summary]**
>
> Again, thanks for all your detailed and valuable suggestions! They are really helpful and have guided us to strengthen the paper. We hope that our revision and comments have resolved your concerns!

---

> > ### Comment · Reviewer_925W · 2022-11-30
> > **Paper clarity improved. Question on controllability and impact on other TTS setups?**
> >
> > Thank you for all the responses, and for updating the paper plus demo page with more details. They’ve addressed my questions on k-means, the dependency structure between style, speaker, text, prosody, and duration, and how the approach performs on unseen text.
> >
> > Thank you also for elaborating on the description of prosody, and the attributes involved, which leads to the question of what it means for prosody to be controllable in TTS? How does this work relate to the literature on controllability of prosody in TTS? This work provides example-based control and interpolations, while [1], [2], etc enable users to edit the pitch contour, or control speaking rate, pitch variation, and affect control to control prosody synthesis. How might this work complement or enhance the others?
> >
> > Another question: Does the amount of supervision (i.e. paired, aligned text audio data) impact the results? Would this approach be helpful for TTS systems that use a lot more supervision?

---

> > > ### Author Response · Authors · 2022-11-30
> > > **Reply from the authors**
> > >
> > > Thank you for your reply and the questions. They are really insightful! This is an updated version of our reply incorporating input from another author.
> > >
> > >
> > > **[What does controllability of prosody in TTS mean here?]**
> > >
> > >
> > > As you point out, there are multiple ways to control prosody in TTS. Here our vision of controllable TTS is in terms of a high-level factor like speaking style.  Other examples include dialect density and degrees of emotion.
> > > Or, we want a system that can synthesize the voice of a "50-year-old general" and a "15-year-old soldier" without specialized training data recorded by professional voice actors.
> > > Controlling prosody via manipulating F0, energy, and duration directly, as you suggest, allows for fine-grained manipulation, but it has the challenge that these cues are highly interdependent and good results require a lot of trial and error.
> > > Another control strategy is to use symbolic markers for phrasing and emphasis, which **implicitly characterize the interdependence of these acoustic correlates**.
> > > In principle, the symbolic control strategy could be implemented with prosodyBERT by learning a mapping with supervision similar to the approach used for predicting unsupervised phonetic units, which could provide **a good alternative to direct control of F0, energy and duration features for word-level control of prosody**.
> > > We can mention this as a possibility for future work.
> > > The initial results in this paper show that vector-based interpolation and control with ProsodyBERT have a strong potential for TTS, and the advances provide a basis for exploring other control strategies.
> > >
> > >
> > > **[Does the amount of supervision impact the results?]**
> > >
> > >
> > > In this work, we haven't done a detailed ablation studies on that, but in principle, more supervision will help improve the results.
> > > The prosody predictor, duration predictor, and the forced alignment to unsupervised alignment (FA2UA) module in UTTS are all trained with aligned text audio data, and more supervision will give better predictions. Similarly, we believe that this approach will be helpful for TTS systems with a lot more supervision. Due to the dataset limitation, we can only demonstrate two styles in this paper. A good follow-up work would be using ProsodyBERT to achieve fine-grained controllable TTS by training on more annotated data with more styles.
> > >
> > >
> > > **[Summary]**
> > >
> > >
> > > Thank you so much for bringing up these questions! They are really insightful and have led to the bigger vision behind this paper. We really appreciate your effort and time in reviewing our paper, contemplating its implications, and asking such good questions. We are happy to discuss more!

---

### Author Response · Authors · 2022-11-17
**Summary response to all reviewers and the new revision**

We sincerely thank all the reviewers for the very helpful comments, which clarified for us some aspects of our approach that were not explained well in the paper.  In particular, the ProsodyBERT model is proposed as a mechanism to generate continuous vector embeddings, not a quantized representation of prosody.  As in HuBERT, K-means just provides the inductive bias for self-supervised learning to derive vector embeddings in the absence of word transcriptions. For TTS, we now realize that the figures were confusing, and have revised these as well as adding more details about the methods. We have substantially revised the paper to address these and other concerns, as described in more detail in specific responses to the reviewers below. Due to the need for adding material and the page limits, we have moved sample spectrograms and some implementation details to the appendix.


The updates are summarized as follows:
* Substantial revisions on the Introduction, Related Work, TTS, and Results sections.
* More demos on unseen text and expressiveness control
* Discussion about the baselines in Section 5.2
* Quantitative results on the duration/rhythm prediction in Section 6.2 and Appendix B, which show that ProsodyBERT features implicitly encode duration information.

---

### Decision · Program_Chairs · 2023-01-20

**Decision:**

Reject

**Justification For Why Not Higher Score:**

The claims of the paper (even iterated by the authors several times in their replies as well) are not justified by their experiments.

**Justification For Why Not Lower Score:**

N/A

**Metareview: Summary, Strengths And Weaknesses:**

The paper uses low-frequency Mel bins, f0, energy as input to train a HuBERT, and term the model ProsodyBERT. The paper claims the learned representation is disentangled and contains prosody information. To validate the claim, the paper includes experimental results on TTS and emotion recognition.

Overall, the reviewers are not convinced that experimental results support the claim that the representation is disentangled and contains prosody information. The paper does not have a definition of disentanglement, nor does it evaluate disentanglement. The paper also does not evaluate what information is in the representation. An improvement in emotion recognition might be attributed to something else, either in the representation or due to the modification of the system. This is questioned by reviewer 925W for the lack of grounding.

The TTS results are further questioned by reviewer J32w. The system is not a mainstream TTS system, and there are no proper baseline for the TTS system (for example, anchoring on other representation such as the one from HuBERT).

Both reviewers J32w and CWFe are concerned about the design decisions and how they impact the results.

The paper can be improved by having fine-grained probing tasks for evaluating the representation, experimenting on a more mainstream TTS system, and doing careful ablation studies on the design decisions.

Reviewer PaY1 gives a high score, but does not include sufficient evidence why the paper warrants the high score.

**Summary Of Ac-Reviewer Meeting:**

We went through the novelty and weaknesses of the paper.